# Iterative phase contrast CT reconstruction with novel tomographic operator and data-driven prior

**Stefano van Gogh**[1,2]*, **Subhadip Mukherjee**[3], **Jinqiu Xu**[1,2], **Zhentian Wang**[5,6], **Michał Rawlik**[1,2], **Zsuzsanna Varga**[4], **Rima Alaifari**[7], **Carola-Bibiane Schönlieb**[3], **Marco Stampanoni**[1,2]

**1** Department of Electrical Engineering and Information Technology, ETH Zürich and University of Zürich, Zürich, Switzerland, **2** Photon Science Division, Paul Scherrer Institut, Villigen, Switzerland, **3** Department of Applied Mathematics and Theoretical Physics, University of Cambridge, Cambridge, United Kingdom, **4** Institute of Pathology and Molecular Pathology, University Hospital Zürich, Zürich, Switzerland, **5** Department of Engineering Physics, Tsinghua University, Beijing, China, **6** Key Laboratory of Particle and Radiation Imaging of Ministry of Education, Tsinghua University, Beijing, China, **7** Department of Mathematics, ETH Zürich, Zürich, Switzerland

* stefano.van-gogh@psi.ch

**Data Availability Statement:** An in-silico breast phantom along with its simulated sinogram has been uploaded to: https://www.research-collection. ethz.ch/handle/20.500.11850/561602. Scanned

## Abstract

Breast cancer remains the most prevalent malignancy in women in many countries around the world, thus calling for better imaging technologies to improve screening and diagnosis. Grating interferometry (GI)-based phase contrast X-ray CT is a promising technique which could make the transition to clinical practice and improve breast cancer diagnosis by combining the high three-dimensional resolution of conventional CT with higher soft-tissue contrast. Unfortunately though, obtaining high-quality images is challenging. Grating fabrication defects and photon starvation lead to high noise amplitudes in the measured data. Moreover, the highly ill-conditioned differential nature of the GI-CT forward operator renders the inversion from corrupted data even more cumbersome. In this paper, we propose a novel regularized iterative reconstruction algorithm with an improved tomographic operator and a powerful data-driven regularizer to tackle this challenging inverse problem. Our algorithm combines the L-BFGS optimization scheme with a data-driven prior parameterized by a deep neural network. Importantly, we propose a novel regularization strategy to ensure that the trained network is non-expansive, which is critical for the convergence and stability analysis we provide. We empirically show that the proposed method achieves high quality images, both on simulated data as well as on real measurements.

## Introduction

Despite ever-improving diagnostics and therapies, breast cancer remains the most common malignancy in women [1]. To fight this public health burden, better imaging techniques are needed to improve early detection of the disease and increase survival rates. In fact, none of the currently used breast imaging techniques (mammography, breast ultrasound, breast MRI

mastectomy data can be found at: https://www.research-collection.ethz.ch/handle/20.500.11850/559555.

**Funding:** M.S.: - ETH-Research Commission Grant Nr. ETH-12 20-2, https://ethz.ch/de/forschung/research-promotion/eth-grants.html - Promedica Stiftung Chur, no URL - SNF Sinergia Grant Nr. CRSII5 183568, https://www.snf.ch/en/HzVMPWm96mz69ZJ8/funding/programmes/sinergia - Swisslos Lottery Fund of Kanton Aargau, https://www.swisslos.ch/de/informationen/guter-zweck/kantonale-fonds/funktion-und-adressen.html S.v.G.: - ETH Doc.Mobility Fellowship The funders had no role in study design, data collection and analysis, decision to publish, or preparation of the manuscript.

**Competing interests:** The authors have declared that no competing interests exist.

and absorption-based breast CT and tomosynthesis [2, 3]) is able to deliver fully three-dimensional images with sufficiently high isotropic resolution and soft-tissue contrast necessary to identify critical breast cancer imaging biomarkers [4]. X-ray phase contrast CT could potentially offer a solution by combining higher soft tissue contrast with the high three-dimensional resolution which characterizes CT [5].

When X-ray waves interact with matter, their amplitude and phase are modified according to the refractive index of the material they interact with. In particular, the refractive index of a given material is described as $n = 1 - \delta + i\beta$. The real part of the index of refraction $\delta$ determines the change in the wavefront's phase $\Phi$ as

$$\Phi = \int \delta \mathrm{d}l, \tag{1}$$

where $l$ indicates the direction of the X-ray beam. From this, the refraction angle $\alpha$ can then be calculated using

$$\alpha = \frac{\lambda}{2\pi}\frac{\partial \Phi}{\partial x}, \tag{2}$$

where $\lambda$ is the X-ray's wavelength. The imaginary part of the index of refraction $\beta$ determines the attenuation coefficient $\mu$ via $\mu = 4\pi\beta/\lambda$, which can then be used to calculate the beam's attenuation by using the Beer-Lambert law.

It is well known that different soft tissues have similar $\beta$'s [5], which leads to little contrast between different tissue types in conventional CT. On the contrary, these differences are larger in $\delta$'s [5]. Consequently, this can theoretically lead to higher soft tissue contrast in the reconstructed CT volumes [6].

X-ray detectors cannot directly measure the X-ray's phase. Many techniques have been proposed over the years to solve this problem: propagation-based [7], crystal interferometry [8], analyzer-based [9], edge-illumination [10] and grating interferometry [11–13]. The latter is a promising approach for in-vivo imaging since it satisfies the prerequisites for clinical compatibility: it has non-restrictive requirements in terms of temporal and spatial coherence of the X-ray beam, it can be operated at large fields-of-view (FOV) and it has a comparably high mechanical robustness [11].

Talbot-Lau grating interferometry detects the X-ray's refraction angle $\alpha$ by exploiting a particular interference pattern called Talbot carpet [14]. When an X-ray beam is refracted, this results in a lateral shift in the interference pattern. Therefore, by measuring this shift, the wavefront's change in phase can be easily obtained by integrating Eq (2). To obtain the Talbot carpet, three gratings are placed between the source and the detector [12]. The first grating (source grating or G0) is made of highly absorbing material and is placed right in front of the X-ray source to increase the beam's coherence. The second grating (phase grating or G1), which is not designed to absorb photons, imposes a significant phase shift to the X-ray beam and creates the interference pattern. Typically, the detector has a resolution in the range of hundreds of micrometers and cannot resolve the pattern of a few micrometer period. Therefore, a highly absorbing third grating (analyzer grating or G2) is placed in front of the detector. By moving one of the gratings with respect to the others in the $x$-direction (see (2)), it is then possible to obtain an interferogram called the phase stepping curve [15], from which it is in turn possible to compute the lateral shift of the interference pattern.

The phase stepping curve can be modeled as

$$I_k = I_0 T \cdot (1 + V_0 D \cdot \cos(k + \Phi_0 - \varphi)). \tag{3}$$

The transmission signal $T$, the dark-field signal $D$, and the differential phase signal are obtained with:

$$T = \exp\left(-\int \mu \, \mathrm{d}l\right), \tag{4}$$

$$D = \exp\left(-\int \epsilon \, \mathrm{d}l\right), \tag{5}$$

$$\varphi = \frac{\lambda d_2}{g_2} \frac{\partial}{\partial x} \int \delta \, \mathrm{d}l. \tag{6}$$

Here, $g_2$ is the pitch of the G2 grating and $d_2$ denotes the distance between the origin and G2. $I_0$, $V_0$ and $\Phi_0$ are the corresponding flat-field intensity, visibility and phase maps, respectively, and $k$ is the $k$-th phase step.

By combining the phase stepping curves of a flat-field measurement with the actual scan, it is possible to obtain the interference pattern's shift $\varphi$, i.e. the differential phase contrast (DPC) signal, with simple a Fourier analysis. While with grating interferometry (GI) it is also possible to obtain the absorption signal $\mu$, which is related to the average intensity of the curve, and the dark-field signal $\epsilon$, which is related to the curve's amplitude, in this paper we focus exclusively on the phase $\delta$. If combined with a CT acquisition protocol, GI naturally extends to GI-CT. Note that since grating interferometry detects the first derivative of the phase (see (6)), during reconstruction an integration step is required if phase contrast tomograms are to be reconstructed.

In an attempt to translate this promising technology into clinical practice, our group has embarked in a long term effort to build a first-of-its-kind GI breast CT (GI-BCT) prototype. As such, our scanning protocol must respect clinical constraints in terms of deposited radiation dose, scanning times and patient comfort.

It is very challenging to obtain high quality images with GI-CT when operated in low dose regimes and with fast acquisition protocols. In fact, to date, the successful use of GI phase contrast CT has been limited to synchrotron beamlines [16] and laboratory setups [17, 18] where high image quality is achieved by a high X-ray flux or by long scanning times, respectively.

Since conventional absorption-based CT has been used in the clinics for decades now, a natural question that arises is: "Why is it more challenging to perform fast low-dose GI-based phase contrast CT as compared to fast low-dose absorption contrast CT?"

This question has two main answers, both of which are related to the fact that the phase cannot be measured directly. First of all, there is an intrinsic noise amplification during the signal retrieval of the differential phase in the projection domain [19]; secondly because the differential nature of the phase contrast forward operator causes the inverse problem to be more severely ill-conditioned as compared to conventional CT.

The amount by which the noise gets amplified during signal retrieval depends on the quality of the grating interferometry setup. The dominant characteristic here is the visibility, i.e. the amplitude of the interference pattern in a flat-field scan. The higher the visibility, the more precisely one can compute the interference pattern's lateral shift and, consequently, the X-ray beam's refraction. High visibilities are notoriously challenging to achieve and the problem of noise amplification must thus be addressed by using a powerful denoising (or regularization)

strategy, which is the first challenge this article aims to address. We would like to highlight that while for attenuation denoising is a relatively easy task because of the nature of its noise power spectrum (NPS), for phase it is more challenging since the noise increases with decreasing spatial frequency [6].

The second challenge is more difficult to tackle since the ill-conditioning of the problem is intrinsically coupled to the physics of the signal acquisition in grating interferometry. Nonetheless, the implementation of the forward-backward operator pair has an impact on the stability of the reconstruction process. This article proposes new operators which are less ill-conditioned and thus less prone to instabilities as compared to the mainstream implementation [20].

In light of the aforementioned two challenges, it is important to reconstruct the tomograms with a stable and powerful inversion algorithm. A pseudo-inverse such as the filtered backprojection (FPB) algorithm could be applied in conjunction with the Hilbert filter [21]. However, it is widely accepted in the CT community that iterative reconstruction algorithms are better suited to deal with highly ill-conditioned problems.

In iterative reconstruction a variational loss function is typically defined which comprises a data-fidelity term and a regularization functional that incorporates prior knowledge about the expected reconstruction. Minimization of this loss with an optimizer of choice then allows to reconstruct an image which is simultaneously consistent with the measurements as well as with the prior knowledge.

A widely used prior in image reconstruction is the total variation (TV) prior [22] which promotes sparse edges in the solution by assuming piece-wise constant signals. While TV is still considered to be a powerful baseline algorithm to regularize ill-conditioned inverse problems, recent years have witnessed the rise of data-driven algorithms which outperform traditional methods.

In addition to pure black-box models, two main data-driven approaches that draw inspiration from classical variational optimization schemes have been proposed. The first approach comprises end-to-end methods which unroll iterative schemes, thereby transforming each iteration of the iterative reconstruction algorithm into a distinct layer of a neural network [23, 24]. Since the imaging physics is embedded into the network, these models are generally believed to be more robust to noise and adversarial perturbations compared to pure black-box neural networks and more data efficient too. In the second approach, the idea is to learn the regularizer a-priori on a representative training set, and to then use the trained regularizer in conjunction with the data-fidelity term in a classical variational optimization framework [25–31].

The first type of approach tends to yield superior results and delivers faster reconstructions [24]. However, it has the significant disadvantage that no convergence guarantees can be derived and that it needs explicit supervision. Algorithms in the second category tend to be slower and potentially yield slightly inferior results. However, they have several advantages. The algorithms can be trained independently of the forward operator, they are more data-efficient (and possibly unsupervised), and they are amenable to stability and convergence analysis [27]. Since it is of utmost importance in the medical field to reliably reconstruct the tomograms, we propose a novel algorithm that fits into the second category.

## Contributions

To address the two major challenges in GI-BCT outlined above and thereby be able to reconstruct phase contrast tomograms from noisy measurement data, we propose an iterative reconstruction algorithm which 1) uses a less ill-conditioned forward-backward tomographic

operator pair, and 2) leverages the power of deep learning to regularize the (still) highly ill-conditioned tomographic inversion problem. In particular, we propose an algorithm that alternates between data updates governed by the L-BFGS algorithm [32] and regularization steps performed with a deep neural network in a Plug-and-Play fashion [33]. Furthermore, we propose an efficient spectral norm regularization strategy which enforces the network to be globally non-expansive.

We apply the proposed approach to both simulated data and real measurements and show that it achieves excellent results. Importantly, we show that it is possible to train the regularizer on simulated data and later apply it to real measurements. This is an extremely useful characteristic since training data is always scare in a medical context.

We would like to mention that while to the best of our knowledge this paper is the first one to combine phase contrast CT reconstruction with a data-driven prior, many articles have been published in the past on iterative phase contrast CT reconstruction [34–37]. Likewise, many works have investigated the synergy of deep learning and iterative methods. Romano et al. introduced the concept of regularization by denoising (RED) in which a denoiser is used in alternation with data updates [25]. This concept is similar to the idea of Plug-and-Play regularization [33] and has then later been implemented with neural networks by [28]. A different approach has been taken by Lunz et al. who proposed a data-driven adversarial regularization approach in which the prior is trained on unpaired data to distinguish between "good", artefact- and noise-free images, and "bad" images containing artefacts and/or noise [26], while Mukherjee et al. later proposed a convex variant of the former approach [27]. On the other hand, also generative approaches towards learning a prior have been investigated. Two notable ones include score-based models in which learn the the gradient of the log-likelihood of the distribution [31], and generative adversarial networks which learn the prior distribution by letting a discriminator and a generator network compete with each other [38].

Our work adds an important contribution to the field since, to the best of our knowledge, it is the first one to combine the quasi-Newton optimizer L-BFGS with a data-driven non-expansive Plug-and-Play prior. Our method is especially useful for highly ill-conditioned inverse problems where a quasi-Newton method greatly accelerates convergence compared to simpler first-order methods.

## Materials and methods

### GI-BCT prototype and scanned sample

Our GI-BCT prototype is a 5th Talbot order 3-gratings symmetric interferometer with a pitch of 4.2 μm. A Comet MXR-225HP/11 tube operated at 70kVp and 10mA was used together with a CdTe photon-counting Dectris prototype detector with a pixel size of 75 μm operated at 10Hz.

To demonstrate the effectivenss of the proposed method on real data, we scanned a formalin-fixed mastectomy specimen from an autopsy without any gross pathological changes obtained at the Department of Pathology and Molecular Pathology, University Hospital Zürich. The data has been obtained with written informed consent under the ethical approval KEK-2012_554 obtained from the Cantonal Ethics Commission of canton Zürich. We acquired 600 projections under continuous circular rotation. After a full rotation, we phase-stepped G0. We combined 5 rotations to construct the sinograms sampling the phase stepping curve. A total of 10 scans have been averaged to partially compensate for the low visibility (thus high noise amplitudes) we are currently working with. Future improvements in grating fabrication are expected to make the latter step obsolete.

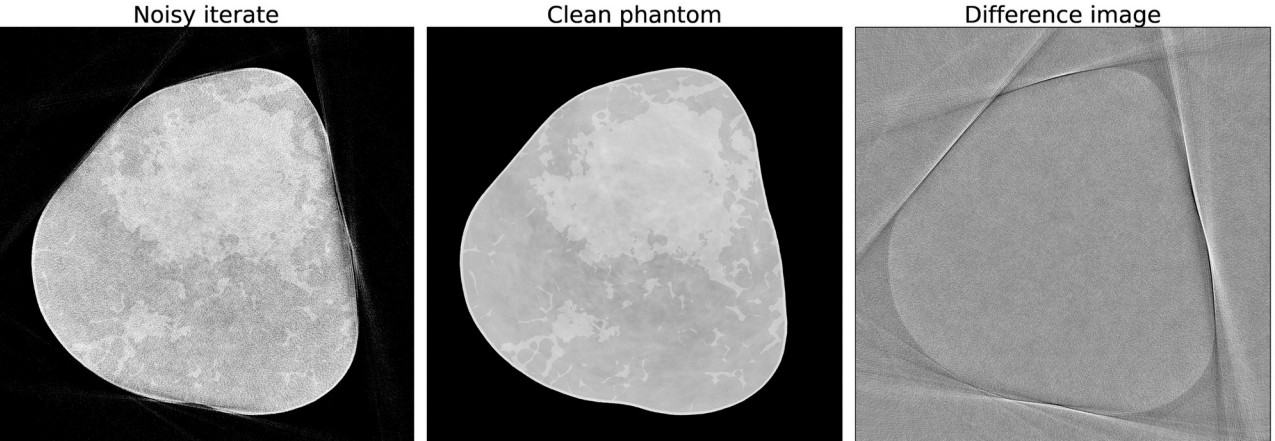

**Fig 1. Training data for the denoising network.** On the left: a noisy iterate obtained after applying 15 gradient updates based on noisy sinogram data; in the middle: a clean breast phantom; on the right: the difference image.

## Simulated data

We used in-silico breast phantoms as described in [39]. DPC sinograms have then been simulated by forward projecting clean phantom data, adding realistic Poisson noise and finally retrieving the differential phase contrast sinogram with Fourier analysis [39]. We simulated 120000 photons leaving the source to get as close as possible to the best data quality we can achieve on our prototype. These DPC sinograms then served a double purpose: 1) to generate training and validation data to train the denoising network, and 2) to quantify the reconstruction performance of the proposed data-driven iterative algorithm.

To generate training and validation data for the denoising network, we ran a few iterations of the (unregularized) optimization scheme by starting from the clean phantom as the initial guess. Thereby, realistic phase wrapping artifacts and noise are introduced in the images, as shown in Fig 1. The clean phantoms and the generated images were then used for supervised training of the network in a 2D slice-by-slice manner. A total of 560 images were used for training and validation.

## DPC forward and backward tomographic operators

The optimization problem we aim to solve is

$$\delta^* = \arg\min_{\delta \in \mathcal{M}} \frac{1}{2} \parallel A\delta - \varphi \parallel_2^2, \tag{7}$$

where $\delta$ is the tomographic volume of the real part of the index of refraction belonging to the set $\mathcal{M}$ of clean phase contrast CT slices, $\varphi$ is the retrieved DPC sinogram and $A$ is a linear forward operator modeling (6). We would like to note that our loss function assumes a constant variance within the retrieved DPC data. While it is possible to compute the expected variance in the data based on the noise model developed in [19], this model requires the true attenuation and dark-field signals as input. In particular, the variance depends quadratically on the (noiseless) dark-field signal induced by the sample. Since it is difficult to accurately estimate the dark-field signal from highly noisy projection data, this uncertainty in the noise model input would lead to large errors in the DPC variance estimation. Therefore, we decided not to incorporate this into our reconstruction pipeline.

In order to iteratively optimize (7) with a gradient-based method, we need to model the forward operator $A$ as well as its adjoint, the backward operator $A^\top$. As shown in (6), the forward operator in grating interferometry's phase contrast channel includes a differential term. While this sets it apart from attenuation-based CT, it remains a linear operator, which is an important prerequisite for analyzing convergence of the proposed algorithm as it ensures convexity of the variational loss.

Two main approaches have been proposed to compute this derivative. In both cases, the differentiation step in the direction perpendicular to the gratings follows the computation of the line integral. The approaches differ in how they compute this derivative. One approach is to numerically approximate it, e.g. by computing a finite difference approximation [40] or by using splines [37]. The other approach parameterizes the image pixels with spherically symmetric expansion functions which allow to analytically obtain the derivative [35]. By computing this derivative a-priori and storing it in a lookup table, it is then possible to sample from this function in the detector plane, thereby circumventing the need for numerical differentiation.

The second approach is more appealing since analytical differentiation is known to be more stable than numerical differentiation, and thus a higher robustness to noisy measurement data can be expected. However, the method proposed in [35] has the disadvantage that it uses a voxel-driven forward model. These operators are known to be non-parallel, and thus less scalable since it is not possible to compute every detector pixel value independently. Ray-driven forward operators on the other hand, are parallel and are thus better suited for large-scale imaging problems. For the inverse operator the opposite is true: voxel-driven methods are easier to parallelize than ray-driven methods.

In this paper, we adapt the method proposed in [35] in order to obtain a ray-driven forward operator and a voxel-driven backward operator, thereby achieving higher scalability. Both operators have been implemented in CUDA for 3D cone beam geometry.

**Forward operator.** What sets the proposed operator apart from the one in [35] is that we sample from the analytical derivative *before* computing the line integrals. In other words, we first calculate how much each pixel refracts the X-rays, before summing up the individual contributions to obtain the total refraction. Besides offering advantages in terms of computational speed and problem conditioning, the proposed implementation more closely resembles the physical process of refraction compared to a somewhat abstract differentiation step in the projection plane.

The rationale of our forward operator is displayed in Fig 2 and described in the following. Let $\delta \in \mathbb{R}^{i,j,k}$ be our phase contrast image and $\phi \in \mathbb{R}^{\alpha,m,n}$ our sinogram with $\alpha$ projections, $m$ detector columns and $n$ detector rows. Before applying the forward operator we compute the analytical derivative of the following spherically symmetric expansion function parameterized by the Kaiser-Bessel window [35]:

$$w_{kb}(x) = \frac{b\left(\alpha_{kb}\pi\sqrt{1 - \left(\frac{x}{w}\right)^2}\right)}{b(\alpha_{kb}\pi)} \tag{8}$$

with

$$b(x) = \sum_{m=0}^{4} \frac{1}{m!(m+2)!}\left(\frac{x}{2}\right)^{2m+3}, \tag{9}$$

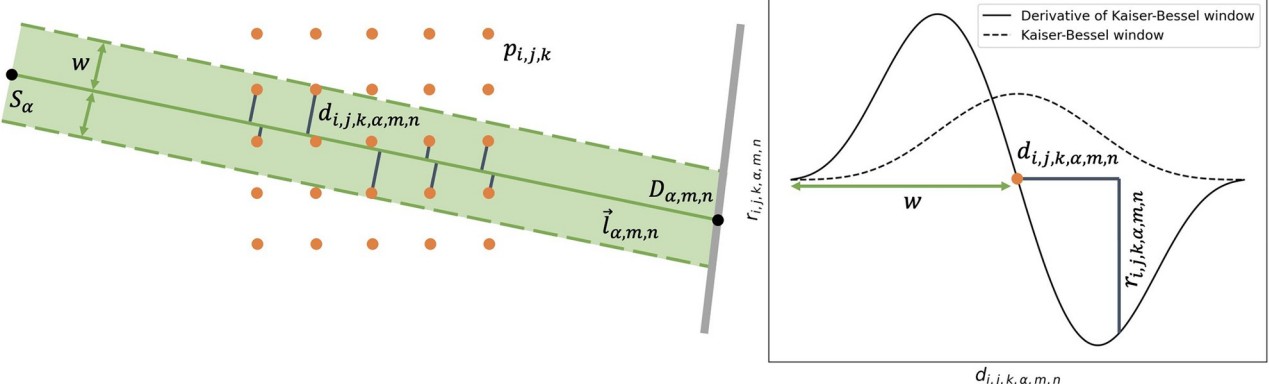

**Fig 2. Ray-driven forward operator for phase contrast CT.** Please note that only a 2D view is provided for simplicity. Image voxel centers $p_{i,j,\,k}$ are displayed as orange dots, the ray $\vec{l}_{\alpha,m,n}$ as a green line and the distances $d_{i,j,\,k,\,\alpha,\,m,\,n}$ of the voxel centers $p_{i,j,\,k}$ to the ray $\vec{l}_{\alpha,m,n}$ are shown in grey. Note that only those pixels are considered which lie within a distance of $w$ (the width of the spherically symmetric expansion function) from ray $\vec{l}_{\alpha,m,n}$. The spherically symmetric expansion function is plotted as a dotted line, its derivative as a continuous line.

where we chose $\alpha_{kb} = 3$ and $w$ twice the image resolution as this empirically led to superior results. Once the lookup table is filled, we first compute the rays $\vec{l}_{\alpha,m,n}$ connecting the X-ray source $S_\alpha$ to detector element $D_{\alpha,m,\,n}$. Then, we determine the minimum distance $d_{i,j,\,k,\,\alpha,\,m,\,n}$ between the ray $\vec{l}_{\alpha,m,n}$ and the voxel centers $p_{i,j,\,k}$. Based on $d_{i,j,\,k,\,\alpha,\,m,\,n}$ we then sample the lateral shift

$$r_{i,j,k,\alpha,m,n} = \frac{\partial w_{kb}}{\partial x}\left(d_{i,j,k,\alpha,m,n}\right) \tag{10}$$

imposed by voxel $p_{i,j,\,k}$ from the derivative of the spherically symmetric expansion function stored in a lookup table. Finally, a sum over all non-zero $r_{i,j,\,k,\,\alpha,\,m,\,n}$ yields the detector count $\varphi$ [$\alpha$, $m$, $n$]. Given the highly parallel nature of this approach, the sinogram $\varphi$ is obtained by computing all its entries in parallel.

Note that the derivative information is considered solely in the horizontal direction. In the vertical direction we used classical linear interpolation to increase computational speed.

**Backward operator.** For the backward operator we follow the reverse logic. Here, every image voxel can be computed independently to yield an highly parallel operator. The concept is illustrated in Fig 3 and explained below.

First, we compute the lines $\vec{v}_{\alpha,i,j,k}$ connecting the X-ray source $S_\alpha$ and the voxel centers $p_{i,j,\,k}$. We then calculate the intersection of line $\vec{v}_{\alpha,i,j,k}$ with the detector plane, thereby obtaining $q_{\alpha,i,\,j,\,k}$. Computing the distance between detector element centers $D_{\alpha,m,\,n}$ and $q_{\alpha,i,\,j,\,k}$, we obtain $t_{i,j,\,k,\,\alpha,\,m,\,n}$, which allows us to sample from the lookup table to get the lateral shift

$$b_{i,j,k,\alpha,m,n} = \frac{\partial w_{kb}}{\partial x}\left(t_{i,j,k,\alpha,m,n}\right). \tag{11}$$

Note that, for cone beam geometry, the lookup table used in the backward operator, which is acting in the detector plane, must be scaled to account for the geometric magnification. Finally, a summation over all angles and detector elements yields the voxel $\delta[i,j,k]$.

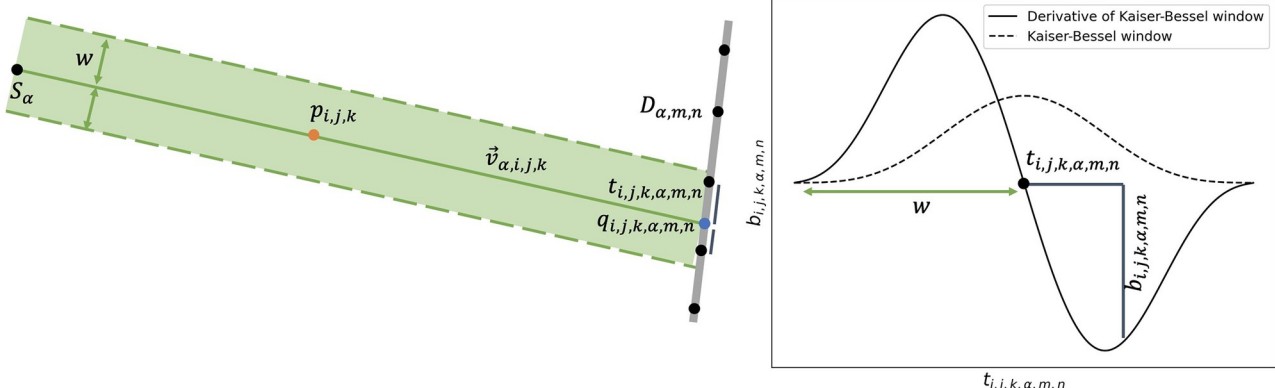

**Fig 3. Voxel-driven backward operator for phase contrast CT.** Please note that only a 2D view is provided for simplicity. One image voxel center $p_{i,j,k}$ is displayed as an orange dot, the line $\vec{v}_{\alpha,i,j,k}$ as a green line. Detector centers $D_{\alpha,m,n}$ are plotted as black dots. The intersection $q_{\alpha,i,j,k}$ of the line with the detector plane is shown as a blue dot. The spherically symmetric expansion function is plotted as a dotted line, its derivative as a continuous line.

In order to save computations, both for the forward and the backward operator, only those voxel centers, respectively detector pixel centers, are considered which lie within a distance $w$ from the traced rays.

**Problem (ill-)conditioning.**   In Fig 4 we show the spectrum of the singular values of the normal matrix $A^\top A$, i.e. the Hessian of (7). We observe that the spectrum of the proposed operator decays at a slower rate compared to the finite difference-based operator. Especially for the larger singular values, our operator is superior. This is somewhat expected since the image has been parameterized by circular expansion functions covering more than one 'classical pixel', which is the parameterization used by the finite difference operator. Therefore, each element in the solution is influenced by a larger number of elements, which increases the curvature of the Hessian in its principal directions, and makes the proposed operator more robust to noise.

Unfortunately, (7) remains a highly ill-conditioned problem even with the new operator. In other words, given the quickly decaying spectrum in Fig 4, the loss in (7) has a very heterogeneous curvature, with some directions being steep, while others being almost flat. The latter makes it difficult to optimize (7) and leads to slow convergence. Moreover, analytically solving (7) yields

$$\delta^* = (A^\top A)^{-1} A^\top \varphi. \tag{12}$$

Given that $A^\top A$ has some very small singular values ($\mathcal{O}(10^{-12})$), a multiplication with its inverse will yield a highly unstable solution. Since an unregularized optimization of (7) will converge to this fixed point, a powerful regularization strategy is necessary to stabilize the reconstruction.

### Data-driven regularizer

As mentioned in the introduction, data-driven regularizers are pushing aside classical regularization strategies such as total variation (TV) [22]. We here propose to learn a regularization network which is able to remove both noise and artifacts from the image iterates in a Plug-and-Play fashion as they converge to the final reconstruction.

Given noisy and clean images as shown in Fig 1, we aim to train a biasless network $f_\theta = W_N R(W_{N-1} \dots R(W_1(x))) : \mathcal{X} \mapsto \mathcal{X}$ with ReLU-nonlinearities $R$, which maps the corrupted image $x$ to its clean counterpart $y$, because for a biasless net there exists an input $x = 0$ s.t.

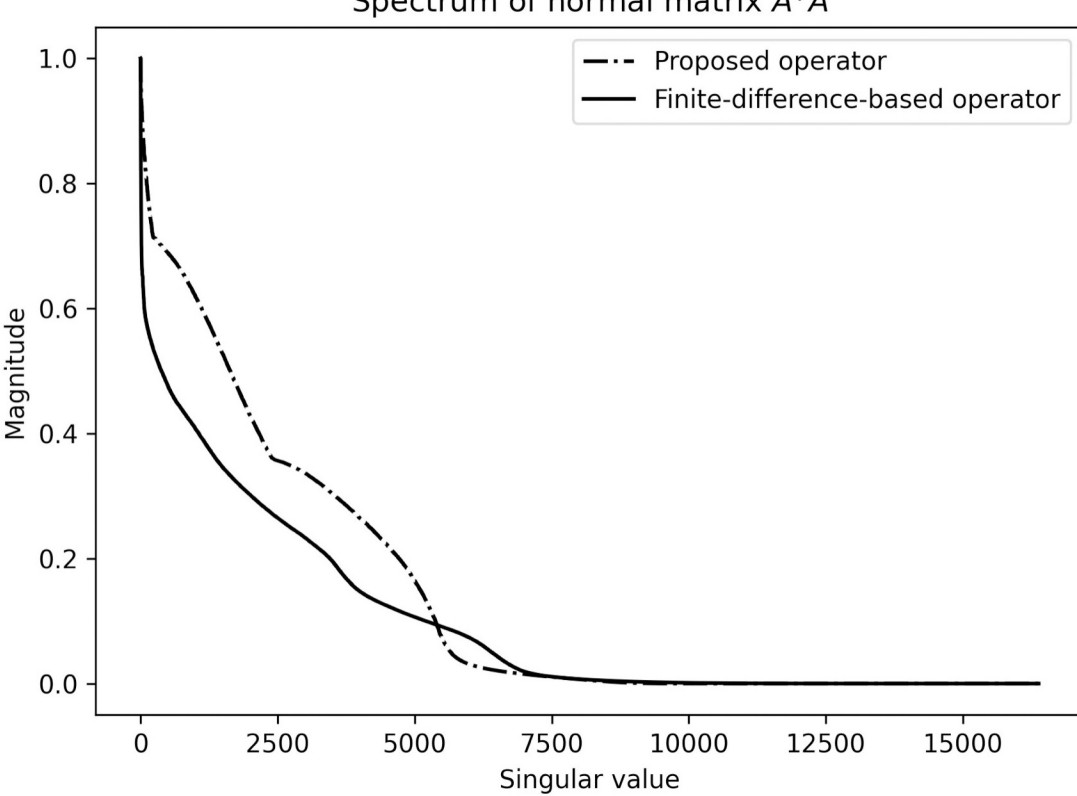

**Fig 4. Spectrum of singular values of the normal matrix $A^T A \in \mathbb{R}^{128^2 \times 128^2}$.** The spectrum of the proposed operator is shown as a dotted line, the spectrum for the finite difference-based operator is displayed as a continuous line. The spectrum of the latter method decreases more rapidly compared to the spectrum of the proposed method. The singular values have been normalized.

$f(x) = 0$, and hence 0 lies in $\mathcal{M}$. This fact will be used to theoretically motivate our algorithm in the Theoretical motivation for PnP-BFGS section. Furthermore, since in the Theoretical motivation for PnP-BFGS section we consider our network to be a projection operator, we aim at ensuring the non-expansiveness of $f_\theta$, i.e. $f_\theta$ should be Lipschitz continuous with Lipschitz constant $L \leq 1$ i.e.

$$\| f_\theta(x_1) - f_\theta(x_2) \|_2 \leq L \| x_1 - x_2 \|_2, \forall x_1, x_2 \in \mathcal{X}^2. \tag{13}$$

Since explicitly constraining the Lipschitz constant of $f_\theta$ to be upper bounded by 1 is an NP-hard problem [41], we penalize the product of the Lipschitz constants of the layers $W_l$ of the network if it is larger than 1. To achieve this we propose the following regularization term:

$$\mathcal{R} = \text{ReLU}\left( \prod_{l=0}^{N} L(W_l) - (1 - \epsilon) \right), \tag{14}$$

with $\epsilon = 10^{-8}$, and where $L(W_l)$ denotes the Lipschitz constant of layer $l$. Since a composition of $L$-Lipschitz functions yields at most an $L$-Lipschitz function, i.e.

$$L(f_\theta) \leq \prod_{l=0}^{N} L(W_l), \tag{15}$$

any minimizer of the regularizer $\mathcal{R}$ ensures that the network $f_\theta$ is non-expansive.

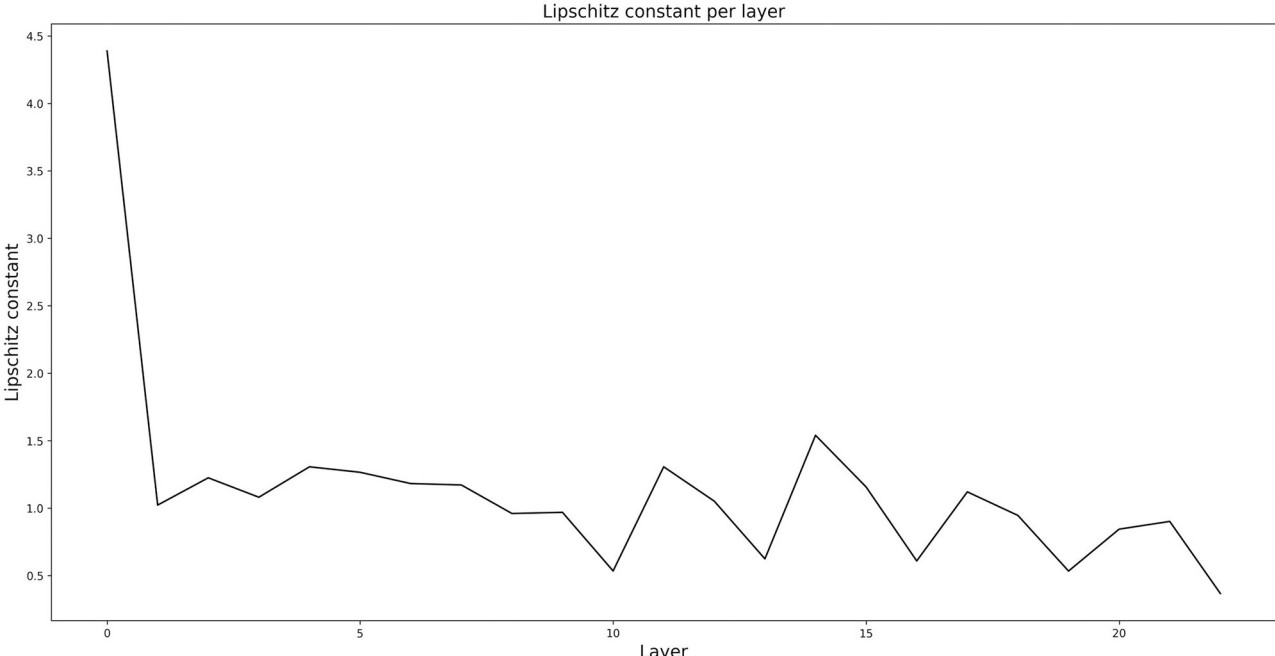

**Fig 5. Lipschitz constants of the network's layers.** The product of the layers' constants is 0.99. This sets an upper bound on the overall network's Lipschitz constant, thus making it non-expansive.

The Lipschitz constant of a layer $l$ can be obtained by computing the spectral norm of $W_l$. Since the spectral norm of $W_l$ corresponds to the largest singular value $\sigma_{l,max}$ of $W_l$, we can compute it using the power method [42].

Our training loss then combines the classical denoising loss with the proposed regularizer $\mathcal{R}$:

$$\mathcal{L}_\theta = \frac{1}{N} \sum_i \| f_\theta(\delta_i) - \delta_i^* \|_2^2 + \lambda \mathcal{R}(\theta), \tag{16}$$

where $N$ is the number of training samples, the clean image $\delta^*$ is sampled from the manifold $\mathcal{M}$, and the noisy image $\delta$ is simulated as explained above. We set $\lambda = 10^{-4}$ and empirically, a (quasi-)minimum of the regularization term, i.e. $\mathcal{R} < \epsilon$, is always reached during optimization. Note that, while the regularizer $\mathcal{R}$ leads to a network which is 1-Lipschitz, the individual layers may not be non-expansive (see Fig 5). In fact, especially the first layer has a fairly high Lipschitz constant. We would like to mention that allowing some layers to be expansive led to significantly improved denoising performance compared to explicitly constraining every layer to be non-expansive. This seems to suggest that it is important for the network to have the freedom to learn some expansive layers. On the other hand, we observed that without spectral norm regularization the product of the spectral norms of the layers diverged.

We parameterized $f_\theta$ with a 7-million-parameter U-net [43]. The model has been trained with Adam [44] until the validation performance stopped improving. Importantly, we removed all biases because it has been shown that 1) this leads to higher model interpretability, and 2) that biasless networks generalize better to different noise amplitudes [45].

The higher interpretability comes from the fact that, in the absence of biases, we can regard the denoising process as being locally linear [45]. Therefore, the Jacobian of the trained network shows how the pixel neighborhood is used to denoise a particular pixel (see Fig 6).

| Noisy | Denoised | Pixel 1 | Pixel 2 |
|---|---|---|---|

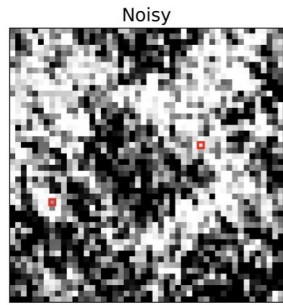 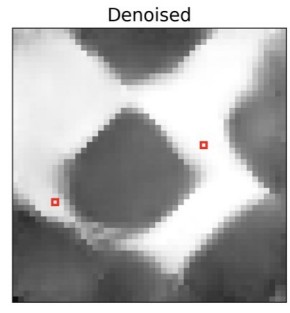 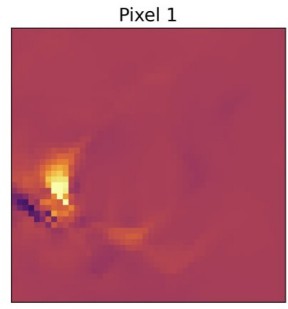 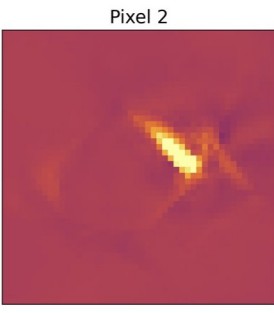

**Fig 6. Model interpretability.** Due to the local linearity of a bias-free network, it is possible to visualize how the denoising happens by computing the Jacobian of the network with respect to the input image [45]. A small image patch is shown for computational reasons. From left to right, we plot the noisy input patch, its denoised counterpart with two pixels highlighted in red. The third and fourth image show how strongly each pixel is weighted to denoise the two pixels highlighted in red.

The high robustness against different amounts of noise is important in our case since 1) the noise encountered during image reconstruction might slightly differ from iteration to iteration, and 2) it makes it easier to train the network as we do not have to find the perfect amount of noise to train on.

Crucially, before inputting the training images into the network, we randomly moved the mean of both the noisy and clean images. This ensures that the network learns to remove artifacts and noise independently of the mean of the image. This is essential in our case because, given the differential nature of the problem, pixels far away from an edge take longer to converge compared to pixels close to an edge. Therefore the local mean of the iterates varies within the images and changes over iterations.

### Iterative phase contrast CT reconstruction with data-driven regularizer

To optimize (7), we propose an alternating optimization scheme which alternates between data updates and denoising steps performed by the data-driven non-expansive denoiser. In particular, the proposed algorithm performs $k$ steps to optimize the data-fidelity term in (7), followed by a denoising step. Given the high ill-conditioned nature of our optimization problem, and the inability of first-order methods to effectively optimize (7), we chose to use the L-BFGS quasi-Newton method [32] to perform the data updates, because of its ability to approximate the Hessian of the loss function, while being very memory efficient.

By removing noise and artifacts, the denoising step with a non-expansive network can be interpreted as an (approximate) projection of the current image iterate $x$ onto the data manifold $\mathcal{M}$ of clean images. Iteratively alternating between data-fidelity optimization and projection to the data manifold thereby allows to approach the measured data, while simultaneously staying within the manifold, and thus yielding a physically relevant solution.

The algorithm starts with an all-zero image and stops when the loss is sufficiently small, i.e. when $\mathcal{L}_k < \epsilon_n$ where $\epsilon_n$ is given by the amount of noise in our data. Since the L-BFGS optimizer restarts with an empty inverse Hessian estimate after each denoising step, our algorithm can be interpreted as a meta algorithm which iteratively applies the L-BFGS method with ever improving starting points.

Both the L-BFGS optimizer and the denoising network have been implemented in Tensorflow 2.0 [46]. The forward and backward operators implemented in CUDA have been registered as Tensorflow functions. The complete algorithm is summarized in Algorithm 1.

**Algorithm 1:** L-BFGS optimization with data-driven Plug-and-Play denoiser

```
input: i = 0; δ₀ = 0; k_max = 15; ε_n;
while ½ || Aδ_k − φ ||₂² > ε_n do
    k = 0; y = 0; s = 0;
    if i > 0 then
        δ_k = δ_reg;
    end
    while k < k_max do
        L_k = ½ || Aδ_k − φ ||₂²;
        ∇_δ_k L_k = Aᵀ(Aδ_k − φ);
            if k > 0 then
                s[k − 1] = δ_k − δ_{k−1};
                y[k − 1] = ∇_δ_k L_k − ∇_{δ_{k−1}} L_{k−1};
            end
            δ_{k+1} = δ_k − LBFGS(∇_δ_k, s, y);
            k = k + 1;
    end
    i = i + 1;
    δ_reg = f_θ(δ_{k_max});
end
output: δ_reg
```

## Theoretical motivation for PnP-BFGS

The following motivating analysis of the proposed reconstruction algorithm holds for BFGS, i.e. the non-memory efficient version of L-BFGS. The key reason for choosing the L-BFGS algorithm with a PnP denoiser is its potential to recover the underlying true image with fewer iterations as compared to first-order methods (e.g., gradient-descent (GD) with PnP denoising). In this section, we formally show that any quasi-Newton (QN) method that builds a progressively accurate approximation of the inverse Hessian can potentially achieve a better convergence rate, thereby leading to significant saving in computations.

Our objective is to recover the image by solving (7). To motivate the use of PnP-BFGS, we consider the following algorithm that alternates between updating the current iterate along a descent direction $d_k = B_k^{-1} g_k$, followed by a projection step onto the image manifold $\mathcal{M}$:

1. $z_{k+1} = \delta_k - \eta_k B_k^{-1} g_k$

2. $\delta_{k+1} = P_M(z_{k+1})$.

Here, $g_k = \nabla f(\delta_k) = A^\top(A\delta_k - \varphi)$ is the gradient of the quadratic loss in (7) evaluated at the $k^{\text{th}}$ iterate $\delta_k$, and $B_k$ denotes a positive-definite matrix that is updated recursively using a suitable QN strategy (which is L-BFGS in our case). To make the analysis simpler, we make the assumption that the data is noise-free, i.e., $\varphi = A\delta^*$. With this simplifying assumption, we have $g_k = A^\top A(\delta_k - \delta^*) = H(\delta_k - \delta^*)$, where $H$ is used as a shorthand for the Hessian matrix $A^\top A$.

We make the assumption that the image manifold $\mathcal{M}$ is a closed and non-empty set containing the zero image, i.e., $0 \in \mathcal{M}$. The pre-trained PnP denoiser $f_\theta$ can be thought of as an (approximate) projection onto $\mathcal{M}$, since it projects noisy images (that are outside $\mathcal{M}$) onto $\mathcal{M}$. Since the convolutional layers in $f_\theta$ have no bias term, $f_\theta(0) = 0$, which is consistent with the assumption $0 \in \mathcal{M}$.

The cone $\mathcal{C}$ at the true solution $\delta^*$ is defined as

$$\mathcal{C} := \{p \in \mathbb{R}^d : p = \alpha(\delta - \delta^*), \text{ for } \alpha \geq 0, \delta \in \mathcal{M}\}.$$

Suppose, the null-space of $H$ satisfies $\ker(H) \cap \mathcal{C} = \{0\}$, i.e., $\mathcal{C}$ contains no vector in $\ker(H)$ apart from the zero vector. This is akin to the standard set-restricted eigenvalue condition considered in [47].

Under the assumptions made above, if the QN algorithm builds a progressively accurate estimate of the left inverse of $H$ over $\mathcal{C}$, then the proposed PnP-BFGS algorithm can potentially achieve a super-linear convergence rate. More precisely, a super-linear rate of convergence can be achieved if the following holds, as we argue later in this section:

$$\lim_{k \to \infty} \sup_{v \in \mathcal{C} \setminus \{0\}} \frac{\| (I - B_k^{-1}H)v \|_2}{\| v \|_2} = 0. \tag{17}$$

At this point, we state the following facts (see [47] for proofs):

Fact-1: If $\mathcal{M} \subset \mathbb{R}^d$ is a closed set, $P_{\mathcal{M}}(z + v) = P_{\mathcal{M}-\{z\}}(v)$.

Fact-2: Let $\mathcal{C} \subset \mathbb{R}^d$ be a closed cone and let $v \in \mathbb{R}^d$. Then, it holds that $\| P_C(v) \|_2 = \sup_{u \in \mathcal{C} \cap \mathcal{B}^d} u^\top v$, where $\mathcal{B}^d$ is the unit ball in $\mathbb{R}^d$.

Fact-3: Let $\mathcal{D}$ be a closed and nonempty set that contains 0. Let $\mathcal{C}$ be a nonempty and closed cone containing $\mathcal{D}$, i.e., $\mathcal{D} \subset \mathcal{C}$. Then for all $v \in \mathbb{R}^d$, it holds that $\| P_{\mathcal{D}}(v) \|_2 \leq \kappa \| P_{\mathcal{C}}(v) \|_2$, where $\kappa = 1$ if $\mathcal{D}$ is convex and $\kappa = 2$ if $\mathcal{D}$ is non-convex. (Lemma 6.4 in [47]).

Using the results above, the error in the $(k+1)^{\text{th}}$ iteration can be bounded as

$$\begin{aligned}
\| \delta_{k+1} - \delta^* \|_2 &= \| P_{\mathcal{M}}(\delta_k - \eta_k B_k^{-1} g_k) - \delta^* \|_2 \\
&= \| P_{\mathcal{M}-\{\delta^*\}}(\delta_k - \delta^* - \eta_k B_k^{-1} g_k) \|_2 \\
&= \| P_{\mathcal{M}-\{\delta^*\}}(\delta_k - \delta^* - \eta_k B_k^{-1} A^\top A(\delta_k - \delta^*)) \|_2,
\end{aligned} \tag{18}$$

where the second equality is a consequence of Fact-1. Now, using Fact-2 and Fact-3 above and setting $\eta_k = 1$, we have

$$\begin{aligned}
\| \delta_{k+1} - \delta^* \|_2 &\leq 2\| P_{\mathcal{C}}(\delta_k - \delta^* - B_k^{-1}H(\delta_k - \delta^*)) \|_2 \\
&= 2 \sup_{u \in \mathcal{C} \cap \mathcal{B}^d} u^\top(\delta_k - \delta^* - B_k^{-1}H(\delta_k - \delta^*)) \\
&\leq 2\| \delta_k - \delta^* - B_k^{-1}H(\delta_k - \delta^*) \|_2 \\
&= 2\| (I - B_k^{-1}H)(\delta_k - \delta^*) \|_2.
\end{aligned} \tag{19}$$

Now, dividing both sides of (19) by $\|x_k - x^*\|_2$ and taking the limit as $k \to \infty$, we notice that

$$\lim_{k \to \infty} \frac{\| \delta_{k+1} - \delta^* \|_2}{\| \delta_k - \delta^* \|_2} = 0, \tag{20}$$

as a consequence of (17). Therefore, under reasonable assumptions on the image manifold $\mathcal{M}$, (20) indicates that PnP-BFGS can achieve super-linear convergence, leading to significantly faster convergence that GD. Nevertheless, it remains to be shown rigorously that (17) indeed holds for L-BFGS-based updates, and, therefore, the analysis presented here only serves as a motivation behind choosing PnP-LBFGS for reconstruction.

## Results

To benchmark the performance of the proposed method, we compared it to three alternative methods: 1) a classical iterative phase contrast reconstruction algorithm which uses a finite difference-based numerical differentiation strategy for the forward and backward operators along with the well-known TV-based regularization [20]; 2) a filtered backprojection; and 3) a

filtered backprojection followed by a deep learning-based post-processing step. For the comparison to be fair, we used the same network for the post-processing as we used within the iterative reconstruction. Please note the latter comparison is easily applicable only when an analytical solution exists (and thus not applicable to cases like [48]). If not, two problems might arise: 1) it could be computationally demanding to create a dataset for training a post-processing net; and 2) in highly ill-posed scenarios, an iterative algorithm might not even be able to converge to a sufficiently high-quality image, which would consequently make the denoising nearly impossible.

To quantitatively evaluate the performance of our model, we reconstructed a $32 \times 1280 \times 1280$ voxel volume based on a $32 \times 600 \times 1280$ pixel noisy sinogram randomly selected from our simulated dataset. As seen in the upper image of Fig 7, this data is corrupted by both

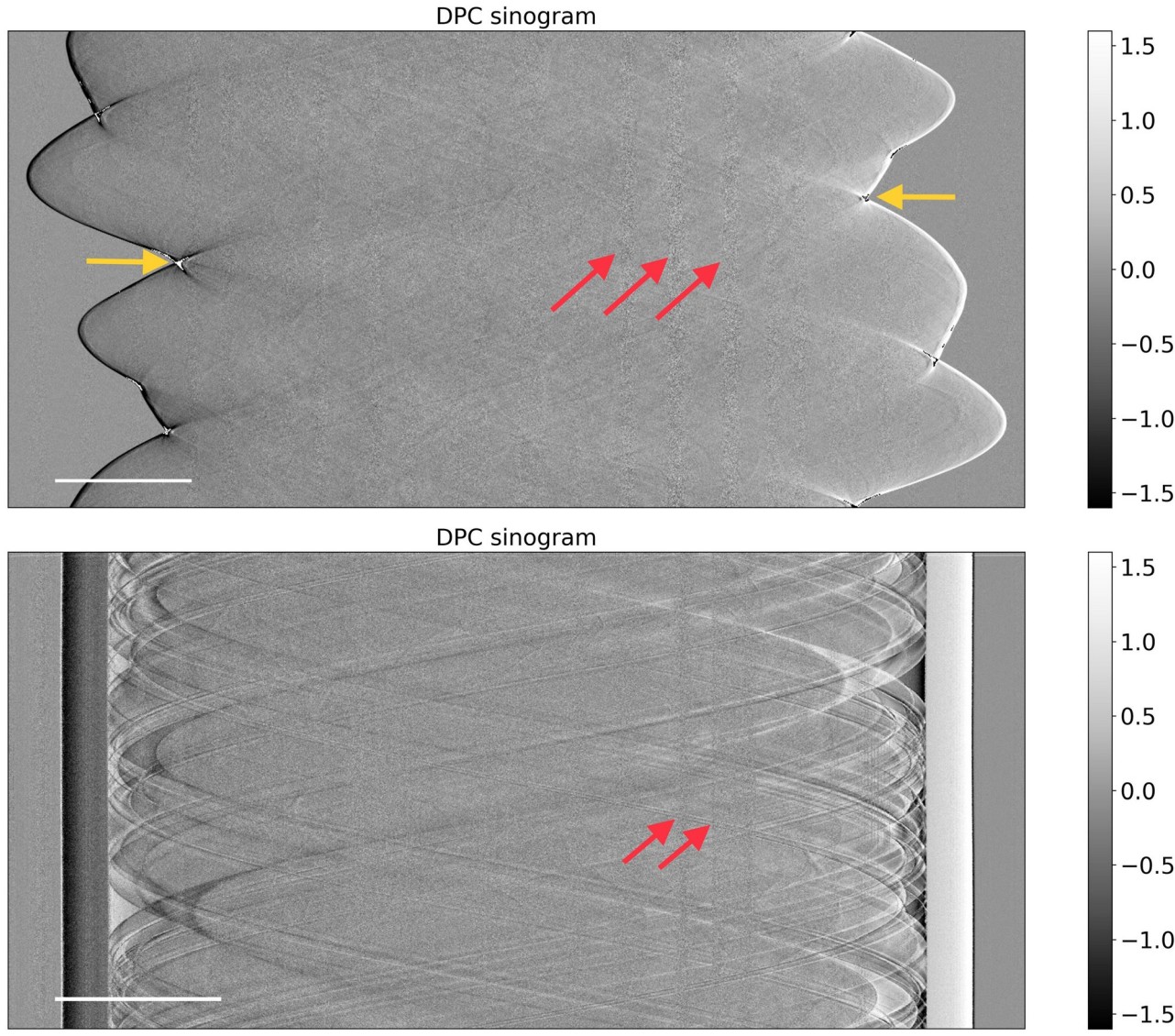

**Fig 7. Simulated sinogram (top) and real sinogram (bottom) used for reconstructing the images in Figs 9 and 10.** The scale bar measures 25 mm. The yellow arrows indicate phase wrapping artefacts, the red arrows indicate localized high amplitude noise.

heteroscedastic noise (see red arrows) as well as by phase wrapping artefacts (see yellow arrows). Both factors need to be compensated for by the regularization functional.

To validate the performance of our algorithm on real measurements we scanned a formalin-fixed mastectomy which yielded the DPC sinogram in the lower part of Fig 7. We reconstructed a $32 \times 1280 \times 1280$ voxel volume based on a $32 \times 600 \times 1280$ pixel sinogram. This data is again corrupted by the heteroscedastic high-amplitude noise highlighted by the red arrows. Phase wrapping artifacts are less pronounced here since there are no straight edges in the scanned mastectomy (see Fig 10).

The computation time for a single data update step of the proposed method was $39s$ on average. The regularization step performed by the trained denoiser had a negligible computational overhead of roughly $100\mu s$. Until convergence the proposed algorithm needed 12 iterations, each composed of 15 data updates combined with one regularization step. Therefore, the whole reconstruction took 117 minutes, i.e. roughly 2 hours. We would like to mention that further optimization of the CUDA implementation of the forward and backward operators will likely enable faster reconstruction times. The baseline algorithm based on finite-difference-based operators and TV regularization had a total reconstruction time of 175 minutes. Here, the long reconstruction time is caused by the regularization step performed with the Chambolle algorithm [49], which is significantly slower compared to a deep learning-based denoiser. On the contrary, the finite-difference-based operator was ca. 8 times faster compared to the proposed operator, which is to be expected since in the proposed operator the voxels are parameterized by basis functions which span multiple classical cubic voxels.

## Effect of the proposed operators

To compare the effect of the proposed operator and the finite difference-based one within the iterative reconstruction scheme, we performed a simple experiment. By starting from the clean ground truth image, we performed a single gradient update based on noisy sinogram data. Since this operation adds noise to the image by propagating it from the sinogram space to the image space, it can be visualized as the displacement of the image iterate away from the manifold of clean images $\mathcal{M}$. Since the task of the denoiser is to project the iterates back to the very same manifold, it is evident that a smaller displacement will result in an easier task for the denoiser.

The result in Fig 8 shows that the proposed operator introduces less noise after one gradient step compared to the finite difference-based operator. While this may be hard to capture by eye, the gradient update of the proposed operator leads to a peak-signal-to-noise-ratio (PSNR) of 11.08 dB, while the baseline method achieves only 9.60 dB. Since these reconstruction algorithms are iterative in nature, this effect gets amplified as the algorithm proceeds, i.e. as the image iterate moves farther away from the manifold $\mathcal{M}$.

We can thus confirm that the proposed operator has an important advantage over the baseline operator since it eases the task of the denoising network, thereby enabling the algorithm to handle larger amounts of noise.

## Effect of the proposed data-driven regularizer

**Simulated data.** Now that we have shown the advantages of the new operator, we will turn our attention to the benefits of using a data-driven prior over a classical one. Fig 9 shows the reconstruction results of simulated data we obtained with the proposed algorithm, along with the comparisons to the TV-based iterative reconstruction [20], filtered backprojection (FBP) and deep learning-based post-processing. We can see that the proposed algorithm achieves excellent results and significantly outperforms the TV-based reconstruction, both

Proposed operator - PSNR: 11.08dB     Finite difference - PSNR: 9.60     Clean

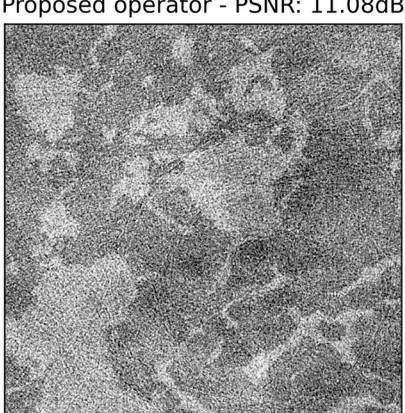 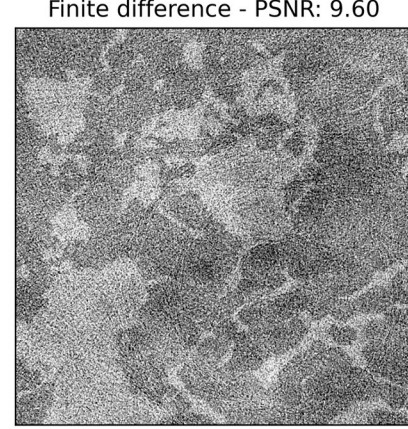 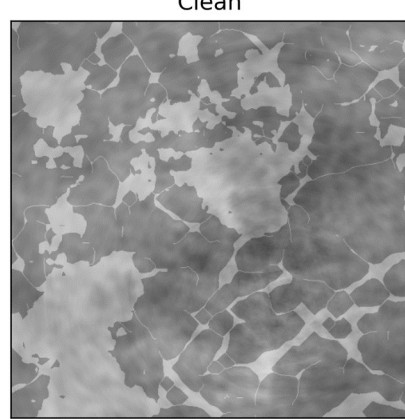

**Fig 8. The effect of one gradient update step.** The proposed operator is shown on the left, the finite difference-based operator in the middle. The clean image used as a starting point is shown for reference (right).

qualitatively as well as in terms of PSNR and SSIM (see Table 1). Moreover, we observe that the proposed method is better at removing phase wrapping artifacts compared to TV-based reconstruction, where some persist at the phantom edge.

A closer look at the TV-based reconstruction reveals the typical staircase artefacts. Interestingly, TV performs better in terms of SNR and CNR. This is attributable to the fact that the latter method promotes piecewise constant signals which translate into low standard deviations and thus higher SNR and CNR values.

A further observation that should be made is that the difference in performance between the TV-based reconstruction and the filtered backprojection is not very large. We believe this is because of the smoothing effect of the Hilbert filter, which suppresses some of the noise, and which is not used when iteratively reconstructing the data.

Finally, a comparison to the deep learning-based post-processing reveals that the latter achieves higher image quality metrics as well as slightly superior visual image quality. This is to be expected since in this case the network has been trained to solve exactly the task it is evaluated on, and it acts on pseudo-inverse solutions of the inverse problem, i.e. FBP with the Hilbert filter. On the contrary, within the iterative reconstruction algorithm, the network acts on intermediate iterates and was not trained to maximize the performance of post-processing, i.e. the network does not get applied to a pseudo-inverse solution of the inverse problem.

**Real data.** We applied exactly the same algorithms as for the simulated data. Importantly, we used the neural networks that have been trained on simulated data, both for the proposed algorithm as well as for the post-processing.

A look at Fig 10 reveals that the proposed algorithm offers superior results compared to TV-based reconstruction also on real measurements, both qualitatively, as well as in terms of SNR and CNR (see Table 1). PSNR and SSIM values could not be calculated since no ground truth image was available. The fact that the denoising network generalizes well to real data, even though it was trained solely on simulations, indicates that our simulations match well with the real data.

The results in Table 1 reveal that, on real data, the post-processing approach achieves an inferior performance compared to the proposed algorithm both in terms of SNR and CNR. A closer look at the images in Fig 10 reveals that there are some artefacts in the post-processed image (see red arrow) which are not present in the proposed method. Given the superior

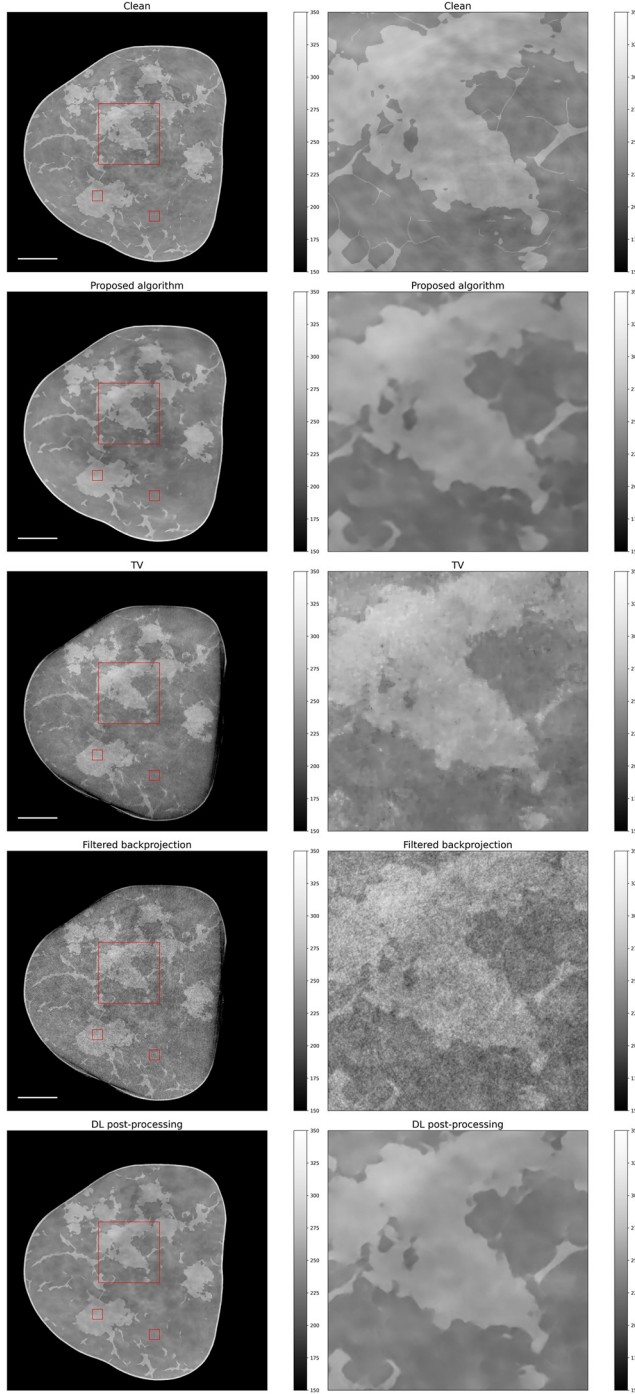

**Fig 9. Reconstruction results on simulated data.** First row: clean phantom; second row: proposed algorithm; third row: TV-regularized algorithm; fourth row: filtered backprojection. On the left a full slice is shown and on the right a zoomed-in section given by the red rectangle is displayed. The white scale bar is 15 mm. The two small red rectangles have been used to compute SNR and CNR values.

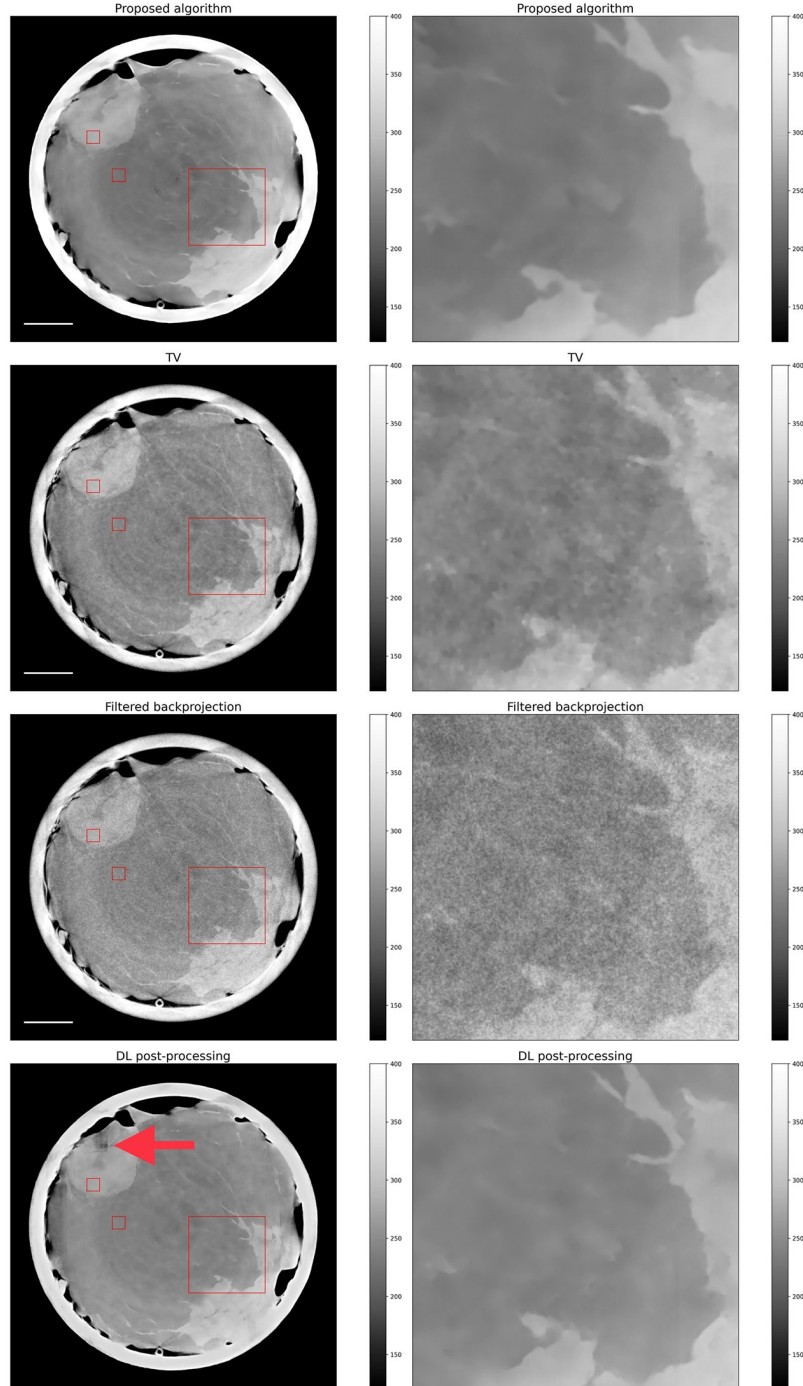

**Fig 10. Reconstruction results on real data.** First row: proposed algorithm; second row: TV-regularized algorithm; third row: filtered backprojection. On the left a full slice is shown and on the right a zoomed-in section given by the red rectangle is displayed. The white scale bar is 15 mm. The two small red rectangles have been used to compute SNR and CNR values.

**Table 1. Quantitative results for the reconstructions in Figs 9 and 10.**

| Simulated data | PSNR | SSIM | SNR | CNR |
|---|---|---|---|---|
| Proposed algorithm | 30.50 | 0.96 | 41.12 | 3.90 |
| TV | 24.73 | 0.91 | 45.35 | 5.71 |
| FBP | 23.05 | 0.77 | 22.30 | 2.09 |
| DL post-processing | 35.32 | 0.97 | 52.57 | 4.42 |
| Real data | PSNR | SSIM | SNR | CNR |
| Proposed algorithm | - | - | 94.80 | 18.38 |
| TV | - | - | 51.78 | 8.70 |
| FBP | - | - | 26.78 | 3.51 |
| DL post-processing | - | - | 77.84 | 9.71 |

performance of post-processing on simulated data, these results suggest that the proposed method might offer higher generalizability. This phenomenon could be explained by the fact that the latter is truly a hybrid approach that takes into consideration the imaging physics and the image prior in a principled way (inspired by quasi-Newton algorithms), whereas a post-processing method does not incorporate the imaging physics.

## Discussion

In this work we have proposed an iterative reconstruction algorithm with novel tomographic operators and a data-driven regularization strategy, along with an efficient strategy to train a non-expansive network.

We could show that the spectrum of the proposed operators, in which the derivative is computed analytically, decays less rapidly compared to the finite difference-based operators, in which the derivative is computed numerically, thus facilitating data-fidelity optimization. Moreover, the new operators propagate less noise, thus easing the task of the denoising network.

The data-driven regularization strategy proved to be significantly superior to TV-based regularization and perhaps more robust in terms of generalization compared to a deep learning-based post-processing step. Importantly, the training loss based on spectral norm regularization allows to efficiently learn a non-expansive network. This approach is applicable to many different fields where a bound on the Lipschitz constant of a network is desired.

In conclusion, we proposed an algorithm which can handle larger amounts of noise compared to the baseline algorithm. Given the noisy data acquired on our setup, the results in this article suggest that a data-driven prior might be indispensable for reconstructing high-quality phase contrast CT images.

## Acknowledgments

The authors would like to thank Dr. Juan Carlos De los Reyes of the Escuela Politécnica Nacional in Quito for the valuable discussions on the L-BFGS optimization algorithm.

## Author Contributions

**Conceptualization:** Stefano van Gogh, Subhadip Mukherjee, Carola-Bibiane Schönlieb.

**Data curation:** Stefano van Gogh, Michał Rawlik, Zsuzsanna Varga.

**Formal analysis:** Stefano van Gogh.

**Funding acquisition:** Stefano van Gogh, Carola-Bibiane Schönlieb, Marco Stampanoni.

**Investigation:** Stefano van Gogh.

**Methodology:** Stefano van Gogh, Subhadip Mukherjee, Michał Rawlik.

**Project administration:** Stefano van Gogh, Marco Stampanoni.

**Software:** Stefano van Gogh, Michał Rawlik.

**Supervision:** Subhadip Mukherjee, Jinqiu Xu, Zhentian Wang, Michał Rawlik, Rima Alaifari, Carola-Bibiane Schönlieb, Marco Stampanoni.

**Validation:** Stefano van Gogh.

**Visualization:** Stefano van Gogh.

**Writing – original draft:** Stefano van Gogh, Subhadip Mukherjee.

**Writing – review & editing:** Stefano van Gogh, Subhadip Mukherjee, Jinqiu Xu, Zhentian Wang, Michał Rawlik, Zsuzsanna Varga, Rima Alaifari, Carola-Bibiane Schönlieb, Marco Stampanoni.

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
