## [Decision Letter · Decision Letter 0]

27 Jun 2022

PONE-D-22-15849Iterative phase contrast CT reconstruction with novel tomographic operator and data-driven priorPLOS ONE

Dear Dr. van Gogh,

Thank you for submitting your manuscript to PLOS ONE. After careful consideration, we feel that it has merit but does not fully meet PLOS ONE’s publication criteria as it currently stands. Therefore, we invite you to submit a revised version of the manuscript that addresses the points raised during the review process.

We look forward to receiving your revised manuscript.

Kind regards,

Jude Hemanth

Academic Editor

PLOS ONE

Journal Requirements:

2. Please amend your Methods section to include the ethics approval number included in your Ethics Statement.

Reviewers' comments:

Reviewer's Responses to Questions

**Comments to the Author**

1. Is the manuscript technically sound, and do the data support the conclusions?

Reviewer #1: Yes

Reviewer #2: Partly

2. Has the statistical analysis been performed appropriately and rigorously? 

Reviewer #1: N/A

Reviewer #2: N/A

3. Have the authors made all data underlying the findings in their manuscript fully available?

Reviewer #1: Yes

Reviewer #2: No

4. Is the manuscript presented in an intelligible fashion and written in standard English?

Reviewer #1: Yes

Reviewer #2: Yes

5. Review Comments to the Author

Reviewer #1: In this work the authors presented an iterative phase contrast CT reconstruction with novel tomographic operator and data-driven prior. The combination of conventional methods and deep learning methods is an important topic. The presented work combines L-BFGS optimization scheme with a deep learning parameterized prior for iterative reconstruction for breast cancer imaging. The theoretical analysis provides the motivation of proposing PnP-LBFGS. The experimental results on simulated and real datasets confirm the advantages of the proposed method.

This reviewer thinks that this paper could be improved in the following ways.

1. The compared baseline methods only include conventional methods. The literature has shown that the deep learning has shown better/competitive performance than iterative algorithms in some applications. Therefore, deep learning-based methods, such as post-processing FBP reconstructed images, should be considered as a competitive baseline.

2. I am wondering if the authors could provide the comparison of reconstruction times, which is a factor for clinical use.

3. In literature there are many works synergizing deep learning and iterative methods. A brief discussion on key references may be important to further highlight the novelty of the proposed methods.

Reviewer #2: The manuscript “Iterative phase contrast CT reconstruction with novel tomographic operator and data-driven prior” discusses an iterative procedure for phase-contrast tomography that make use of a neural network denoising step.

The manuscript is generally well-written and clear, even if many aspects of the proposed strategy are not properly discussed. In addition, often, the authors make statements that are not properly justified or that would need at least some references. About these remarks from my side, more detailed information can be found in the attached pdf.

In general:

1) I am not convinced the proposed approach can be called data-driven as the denoising part is associated with the model space (the images), not the data space;

2) In both the synthetic and the experimental tests, the proposed approach tends to remove many details (that are spatially consistent, so, probably not related to random noise, and that are connecting different subdomains in the reconstruction). My impression is that similar results could be obtained, for example, with a much simpler spatial filter or a more appropriate choice of the regularization within the framework of more "standard" approaches. This, of course, does not mean that the proposed approach is not interesting, but a more accurate and fair comparison with the “mainstream” approaches would make the paper more relevant to the readers.

3) A more in-depth discussion of the noise in the data would be important. Moreover, why no data covariance matrix is included in the inversion algorithm? It seems to me that the proposed strategy might be affected, for example, by severe modeling error that is not taken into account and that might lead to dangerous over-interpretation of the results. I do not expect the authors to modify their algorithm accordingly (at least for this manuscript), but I feel a short discussion about that would be important.

I hope these comments might be helpful in further improving the quality of the paper.

Best

P.S.

Authors must improve the quality of the images.

6. PLOS authors have the option to publish the peer review history of their article (what does this mean?). If published, this will include your full peer review and any attached files.

Reviewer #1: No

Reviewer #2: No

---

## [Author Response · Author response to Decision Letter 0]

22 Jul 2022

Response to Reviewers:

Please find below our rebuttals to the editor’s and reviewers’ comments. We hope we could satisfactorily address the concerns. We highlighted the changes in red in the text.

Editor’s comments:

Please ensure that your manuscript meets PLOS ONE's style requirements, including those for file naming. The PLOS ONE style templates can be found at https://journals.plos.org/plosone/s/file?id=wjVg/PLOSOne_formatting_sample_main_body.pdf and https://journals.plos.org/plosone/s/file?id=ba62/PLOSOne_formatting_sample_title_authors_affiliations.pdf

- We changed the figure labelling from Fig. to Fig and multiple figure labelling to Figs … and …

- We justified the text

- We changed the author affiliations 

- We added the corresponding author’s initials after the corresponding email address

- We changed the last section’s name from conclusions to discussion 

Please amend your Methods section to include the ethics approval number included in your Ethics Statement.

- We added the ethics statement 

In your Data Availability statement, you have not specified where the minimal data set underlying the results described in your manuscript can be found. PLOS defines a study's minimal data set as the underlying data used to reach the conclusions drawn in the manuscript and any additional data required to replicate the reported study findings in their entirety. All PLOS journals require that the minimal data set be made fully available. For more information about our data policy, please see http://journals.plos.org/plosone/s/data-availability. Upon re-submitting your revised manuscript, please upload your study’s minimal underlying data set as either Supporting Information files or to a stable, public repository and include the relevant URLs, DOIs, or accession numbers within your revised cover letter. For a list of acceptable repositories, please see http://journals.plos.org/plosone/s/data-availability#loc-recommended-repositories. Any potentially identifying patient information must be fully anonymized.

- Due to legal restrictions, we are not allowed to share the patient data obtained with written informed consent under the ethical approval KEK-2012 554 granted by the Cantonal Ethics Commission of canton Zürich. We can however share our in-silico breast phantoms and the corresponding sinogram data. They will be uploaded to the ETH research collection archive as supporting material to our paper: https://www.research-collection.ethz.ch. The exact URL will follow.

Reviewers’ comments:

The compared baseline methods only include conventional methods. The literature has shown that the deep learning has shown better/competitive performance than iterative algorithms in some applications. Therefore, deep learning-based methods, such as post-processing FBP reconstructed images, should be considered as a competitive baseline.

- We added the comparison to a deep learning-based post-processing step in the “Results” section and in the “Effect of the proposed data-driven regularizer” section. For the comparison to be fair, we used the same network as within the iterative reconstruction. We would like to stress that this comparison only makes sense when it is possible to analytically reconstruct the phase contrast volumes (which is not always the case, e.g. in Teuffenbach et al., 2017).

- We accordingly edited Figs 9 and 10.

I am wondering if the authors could provide the comparison of reconstruction times, which is a factor for clinical use.

- We added a paragraph in the “Results” section describing the overall reconstruction times of the two iterative methods, as well as a more detailed description of the computational times of the single steps of the algorithms.

In literature there are many works synergizing deep learning and iterative methods. A brief discussion on key references may be important to further highlight the novelty of the proposed methods.

- We added a paragraph in the “Contributions” section that discusses prior work on combining deep learning with iterative methods and explained where the novelty of our method lies and why it is relevant to the scientific community.

I am not convinced the proposed approach can be called data-driven as the denoising part is associated with the model space (the images), not the data space;

- In the machine learning community, the term “data-driven” refers to algorithms that are fitted on training data, to differentiate them from algorithms that are solely designed by human engineering, and which do not contain trainable parameters. The fact that the mapping is applied in image space does not preclude the term data-driven in our opinion.

In both the synthetic and the experimental tests, the proposed approach tends to remove many details (that are spatially consistent, so, probably not related to random noise, and that are connecting different subdomains in the reconstruction). My impression is that similar results could be obtained, for example, with a much simpler spatial filter or a more appropriate choice of the regularization within the framework of more "standard" approaches. This, of course, does not mean that the proposed approach is not interesting, but a more accurate and fair comparison with the “mainstream” approaches would make the paper more relevant to the readers.

- The article already contains a comparison to the most celebrated and successful classical regularization scheme for CT, i.e. TV regularization. As it can be seen from the results in Figs 9 and 10 and in Table 1, TV regularization is not able to achieve comparable performance to the data-driven PnP strategy.

- Moreover, the new comparison with FBP-denoising uses a CNN that is composed of spatial filters, which can thus be regarded as a spatial filter.

- It is true that some small details are lost during the denoising. However, given the high amount of noise in the data and the very small size of the structures, we believe it is unrealistic to hope for those small features to be recovered.

A more in-depth discussion of the noise in the data would be important. Moreover, why no data covariance matrix is included in the inversion algorithm? It seems to me that the proposed strategy might be affected, for example, by severe modeling error that is not taken into account and that might lead to dangerous over-interpretation of the results. I do not expect the authors to modify their algorithm accordingly (at least for this manuscript), but I feel a short discussion about that would be important.

- We added a paragraph in the “DPC forward and backward tomographic operators” section in which we argue why we did assume constant variance in the data. The reason is that to accurately estimate the DPC variance, one needs to know the dark-field signal. Since the dark-field signal is hard to accurately compute based on highly noisy data, we decided not to include this into our model. We are currently working on a new algorithm which explicitly models the variance in the intensity data instead of the retrieved DPC data.

P.S. Authors must improve the quality of the images.

- All images were very high-resolution and all passed the PACE system test. Please let us know in case we must edit them.

---

## [Decision Letter · Decision Letter 1]

1 Aug 2022

Iterative phase contrast CT reconstruction with novel tomographic operator and data-driven prior

PONE-D-22-15849R1

Dear Dr. Gogh

We’re pleased to inform you that your manuscript has been judged scientifically suitable for publication and will be formally accepted for publication once it meets all outstanding technical requirements.

Kind regards,

Jude Hemanth

Academic Editor

PLOS ONE

Additional Editor Comments (optional):

Reviewers' comments:

Reviewer's Responses to Questions

**Comments to the Author**

1. If the authors have adequately addressed your comments raised in a previous round of review and you feel that this manuscript is now acceptable for publication, you may indicate that here to bypass the “Comments to the Author” section, enter your conflict of interest statement in the “Confidential to Editor” section, and submit your "Accept" recommendation.

Reviewer #1: All comments have been addressed

Reviewer #2: All comments have been addressed

2. Is the manuscript technically sound, and do the data support the conclusions?

Reviewer #1: (No Response)

Reviewer #2: Partly

3. Has the statistical analysis been performed appropriately and rigorously? 

Reviewer #1: (No Response)

Reviewer #2: N/A

4. Have the authors made all data underlying the findings in their manuscript fully available?

Reviewer #1: (No Response)

Reviewer #2: No

5. Is the manuscript presented in an intelligible fashion and written in standard English?

Reviewer #1: (No Response)

Reviewer #2: Yes

6. Review Comments to the Author

Reviewer #1: (No Response)

Reviewer #2: (No Response)

7. PLOS authors have the option to publish the peer review history of their article (what does this mean?). If published, this will include your full peer review and any attached files.

Reviewer #1: No

Reviewer #2: No

---

## [Editor Report · Acceptance letter]

16 Aug 2022

PONE-D-22-15849R1 

Iterative phase contrast CT reconstruction with novel tomographic operator and data-driven prior 

Dear Dr. van Gogh:

I'm pleased to inform you that your manuscript has been deemed suitable for publication in PLOS ONE. Congratulations! Your manuscript is now with our production department. 

Kind regards, 

on behalf of

Dr. Jude Hemanth 

Academic Editor

PLOS ONE